# Specific Focus on Antifungal Peptides against Azole Resistant *Aspergillus fumigatus*: Current Status, Challenges, and Future Perspectives

**DOI:** 10.3390/jof9010042

**Published:** 2022-12-27

**Authors:** Dilan Andrés Pimienta, Freiser Eceomo Cruz Mosquera, Isabella Palacios Velasco, María Giraldo Rodas, Jose Oñate-Garzón, Yamil Liscano

**Affiliations:** 1Grupo de Investigación en Salud Integral (GISI), Facultad de Salud, Universidad Santiago de Cali, Cali 5183000, Colombia; 2Grupo de Investigación en Química y Biotecnología (QUIBIO), Facultad de Ciencias Básicas, Universidad Santiago de Cali, Cali 760035, Colombia

**Keywords:** antifungal peptides, *Aspergillus*, antifungal resistance, *Aspergillus fumigatus*, allergic bronchopulmonary aspergillosis (ABPA)

## Abstract

The prevalence of fungal infections is increasing worldwide, especially that of aspergillosis, which previously only affected people with immunosuppression. *Aspergillus fumigatus* can cause allergic bronchopulmonary aspergillosis and endangers public health due to resistance to azole-type antimycotics such as fluconazole. Antifungal peptides are viable alternatives that combat infection by forming pores in membranes through electrostatic interactions with the phospholipids as well as cell death to peptides that inhibit protein synthesis and inhibit cell replication. Engineering antifungal peptides with nanotechnology can enhance the efficacy of these therapeutics at lower doses and reduce immune responses. This manuscript explains how antifungal peptides combat antifungal-resistant aspergillosis and also how rational peptide design with nanotechnology and artificial intelligence can engineer peptides to be a feasible antifungal alternative.

## 1. Introduction

Fungal infections are increasing worldwide and affect people of all age groups, ethnicities, and genders [1,2,3,4,5]. Aspergillosis is a common fungal infection associated with multiple risk factors such as immunosuppression, the use of corticosteroids, pre-existing respiratory tract pathologies (asthma, chronic obstructive pulmonary disease, and bronchiectasis), and the indiscriminate prescription of antifungal drugs [6,7,8,9]. The clinical course of aspergillosis varies across patients prescribed pharmacological management strategies and pathogens; therefore, treatment and prognosis depend on the species of *Aspergillus* spp. in each case [10,11].

Although *Aspergillus* spp. is numerous, *A. fumigatus* is the most frequent etiologic agent and can be attributed to 80% of aspergillosis cases [12,13,14]. Studies have shown that this pathogen can induce a variety of allergic reactions, as well as life-threatening systemic diseases, including allergic bronchopulmonary aspergillosis (ABPA), chronic non-invasive or semi-invasive pulmonary aspergillosis, and invasive airway aspergillosis (API) [15,16,17].

Denning et al. [18] estimated that 4.8 million people worldwide develop ABPA, of whom approximately 400,000 have chronic pulmonary aspergillosis. There were 154,888 cases of API in the United States between 2009 and 2013, which were associated with increased 30-day hospital readmission rates, an excessive length of hospitalization, and costs amounting to USD 600 million annually [19]. In Colombia, about 3000 cases of API occur per year (5.7 cases per 100,000 inhabitants); of which 13% is related to organ transplant recipients and can reach a mortality of more than 70% without adequate antifungal management [20].

The first-line treatment of diseases caused by *A. fumigatus*, particularly API, relies on triazole antifungals, especially voriconazole [21,22]. A randomized controlled clinical trial by Herbrecht et al. [23] compared the efficacy of voriconazole vs. amphotericin B in 379 patients with API and showed greater responsiveness at 12 weeks (54.7% vs. 29.9%, respectively); moreover, survival was markedly superior in subjects treated with voriconazole relative to those using amphotericin B (70.2% vs. 54.9%, respectively).

Despite the efficacy of triazoles, adverse events associated with their chronic use and antifungal resistance have emerged. In a retrospective cohort of 196 patients with API, 37 (19%) had a voriconazole-resistant infection; furthermore, when comparing drug-resistant vs. drug-sensitive cases, an increase in overall mortality of 21% at day 42 was found, reaching approximately 50%. Importantly, voriconazole-resistant variants of *A. fumigatus* frequently exhibit cross-resistance to other agents such as itraconazole, isavuconazole, and posaconazole [24]. This has led to a search for new therapeutic alternatives such as vaccines, nanoparticles, and other therapeutic molecules. Some authors have proposed the use of vaccines with inactivated conidia or live-attenuated conidia to prevent *Aspergillus* spp. disease, or even a universal antifungal vaccine to protect against multiple strains or species [25]. However, these biologics must overcome different host risk factors and the variety of fungal pathologies. On the other hand, in a publication, it was found that the use of zinc oxide nanoparticles (ZnONPs) with an average size of 60 nm and a hexagonal shape had antifungal effects against *T. mentagrophyte, Microsporum canis, C. albicans*, and *Aspergillus fumigatus* [26].

Antifungal peptides have recently emerged as a family of bioactive macromolecules with clinical potential due to their abilities to alter the structures of fungal cells, their broad spectrum of activity, and low resistance response [27]. More than 1000 peptides with antifungal activity have been isolated, mostly of natural origin, derived from plants, animals, and insects, although peptides of semi-synthetic and synthetic origin are gaining attention [28]. Mycosis, β-defensins, lactoferrin, lysozyme, drosomycin, and histones stand out among the antifungal peptides proposed to treat *A. fumigatus*; although variable efficacy has been reported, their essential characteristic is low fungal resistance. Ballard et al. [29] found that lysozyme and histones inhibited hyphal metabolic activity in all *A. fumigatus* isolates tested, and their efficacy increased as a function of dose. In addition, imaging flow cytometry revealed that histones, β-defensin-1, and lactoferrin inhibited the germination of the conidia of this fungus. Similarly, Lupetti et al. [30] found that synthetic peptides derived from human lactoferrin, histatin, and ubiquicidin also attack *A. fumigatus* hyphae in a dose-dependent manner.

Thus, this article reviews the treatment alternatives for the disease caused by *A. fumigatus*, with an emphasis on peptides with antifungal activities against this pathogen.

## 2. Epidemiology and Mechanism of Resistance of *A. fumigatus* to Antifungal Agents

Azole-resistant *A. fumigatus* was first discovered in a clinical setting in 1997 by analyzing isolates collected in the 1980s [31] and has since been widely documented [32,33,34]. This phenomenon is increasingly frequent in patients with no history of recent treatment with azoles and in places where fungicides belonging to this pharmacological family are used for agriculture [35].

An international study reported the prevalence of azole-resistant *A. fumigatus* to be 3.2% in 3788 Aspergillus isolates from 22 centers in 19 countries. Resistance was detected in 11 countries (57.9%), including Austria, Belgium, Denmark, France, Italy, and the United Kingdom [36]. In the Netherlands, azole resistance has an overall prevalence of 5.3%, ranging from 1.8 to 12.8%, depending on the region and hospital. Likewise, specific resistance to itraconazole has increased, fluctuating between 1.7 and 6%. Other countries with more sporadically isolated resistant strains include Spain, Germany, France, China, Iran, and India, where the frequency ranges from 0.3 to 10% [37,38].

In the United States, agricultural azoles can lead to cross-resistance to medical azoles prescribed against *A. fumigatus*, whose prevalence is 2.6%. Hursts et al. [39] developed in an experimental peanut field treated with azole fungicides and found resistant *A. fumigatus* presenting the TR347l98 mutation.

On the other hand, reports of itraconazole resistance in *Aspergillus* spp. from Brazil are scarce. Negri et al. [40] did not observe triazole resistance among 221 clinical isolates of *A. fumigatus* in 2017 in this country. Additionally, a prospective study in Peru included 143 strains of *A. fumigatus* isolated from several hospital centers and reported a prevalence of triazole resistance of 2.09% [41]. In Argentina, Romero et al. [42] found that 8.1% of the *A. fumigatus* isolates studied had decreased susceptibility to itraconazole.

Although there are no robust epidemiological reports of resistance of this pathogen to azoles in countries such as Colombia, the use of fungicides, such as tebuconazole and difenoconazole, has been documented in the flower industry, specifically in Cundinamarca, where 60% of the national production is concentrated. Le Pape et al. [43] found 38 strains of Aspergillus resistant to itraconazole or voriconazole among 60 soil samples from flower fields and greenhouses.

The resistance of *A. fumigatus* to azoles is associated with the mutations of the *cyp51A* and *cyp51B* genes, especially the former in environmental and clinical isolates (see Figure 1) [44]. The *cyp51A* gene encodes the enzyme 14α-esteroldesmethylase, which synthesizes ergosterol, a major component of the fungal membrane [45]. The expression of the *cyp51A* gene encodes the enzyme necessary for mycelial growth, and its deletion of this gene reverses resistance to fluconazole; however, this does not happen when *cyp51B* is deleted. In addition to resistance mediated by *cyp51A* mutations, other resistance mechanisms have been described, such as the overexpression of efflux pumps, adaptation to stress, and resistance by mutations other than *cyp51A* [46].

In the first instance, the overexpression of efflux pumps in *A. fumigatus* allows drug efflux, increasing the likelihood of resistance to azole antifungals. In this regard, one study showed that azole-resistant *A. fumigatus* strains express 30 times more efflux pumps than susceptible strains [47]. This has been associated with the overexpression of genes, such as *AtrF, AfuMDR1, AfuMDR2, AfuMDR3, AfuMDR4*, and *MfsA-C* [48]. On the other hand, *A. fumigatus*, such as other fungi, activates complex signaling pathways to adapt to the hostile environment induced by azole [49]. Studies suggest that the calcium signaling pathway mediates the antifungal activity of azoles against this pathogen; the combination of azoles with calcium inhibitors, thus, increases their efficacy in vitro and in vivo [50,51].

The last mechanism of azole resistance is through mutations other than *cyp51A*, which are seldom detected in azole-resistant clinical isolates despite their identification in vitro. Wei et al. [52] showed that mutations in the *algA* gene encoding the calcium-dependent protein are associated with isolates of *A. fumigatus* resistant to azole antifungals. Hagiwara et al. [53] also reported itraconazole resistance in *A. fumigatus* strains with nonsynonymous mutations of the *afyap1* and *aldA* genes.

## 3. Alternative Therapy

### 3.1. Antifungal Peptides

Given the limitations of the alternatives, peptides are a valuable and viable treatment option for patients infected by *A. fumigatus*. Antimicrobial peptides (AMPs) are molecules composed of 10–50 amino acids (see Table 1), and they are generally cationic due to their high amounts of arginine and lysine; their amphipathic properties can be attributed to a large fraction of hydrophobic amino acids [54]. Although they are genetically encoded [55], their amino acid sequences, sizes, and structures may diverge.

Lima et al. [56] clarified six essential characteristics of AMPs that determine their function: helicity, charge, hydrophobicity, length, sequence, and self-association. Regarding helicity, some authors have reported that the substitution of different amino acids changes the ability of peptides to adopt the correctly folded α-helix structure; the introduction of residues also changes the hydrophobic character of the same [57,58]. On the other hand, although the positive charge of the peptides mediates the initial attraction to the microbial membrane (negatively charged), whether there is a directly proportional relationship between the positive charge and the antimicrobial activity of AMP cannot yet be ascertained [59].

Regarding length, it has been stated that an AMP requires at least 18 residues to cross the cell membrane [56]. Studies performed on peptides such as cathelicidin and HP-A3 showed that reducing their length can decrease their antimicrobial activity seven-fold and even stops hemolytic activity [60,61]. Regarding self-association, it has been shown that increasingly aqueous environments decrease the antimicrobial potential of an AMP [56].

Insect lymph, neutrophil granules, and other cells of the immune system, as well as the skin of some frogs, contain antifungal peptides capable of killing fungi. Therefore, there is a great variety of peptides with antifungal properties across almost all vertebrate species that have developed through duplication, natural selection, and specific pathogens [62].

In nature, AMPs are produced in two ways: by the ribosomal translation of mRNA or by non-ribosomal synthesis. In the first instance, peptides synthesized in ribosomes are genetically encoded in all life forms, including bacteria, and their therapeutic potential can be attributed to their role in innate immunity [63]. On the other hand, non-ribosomal peptides are mRNA-independent, are mainly produced by the secondary metabolism of bacteria and fungi, and have been used for decades as antibiotics [64].

Although more than 1251 AMPs with antifungal activity have been isolated, there are about 31 with activity against *A. fumigatus* according to the AMP database (Table 1).

**Table 1 jof-09-00042-t001:** Antifungal peptides with activity against *A. fumigatus.* The table presents the antifungal peptides of natural origin with their physicochemical parameters and minimum inhibitory concentration against *A. fumigatus*.

Author/Year	Name	Source	Length	Net Charge	Hydrophobic Residues	Boman Index	MIC
Mignone et al., 2022 [65]	Sin a 1	Seeds, white mustard, *Brassica hirta*	145	10	32%	2.09	63 µM
Seyedjavadi et al., 2019 [66]	M. chamomilla AMP 1	*Matricaria chamomilla L.*	23	3	47%	0.68	6.66 µM
Khani et al., 2019 [67]	Skh-AMP1	Leaves, *Satureja khuzistanica*	25	5	28%	3.19	20.7 µM
Xiaoxia et al., 2019 [68]	*P. xylostella Moricin*	Highly expressed in fat body and hemocyte, diamondback moth, *Plutella xylostella*	42	9	35%	1.67	8.9–23 µM
Park et al., 2016 [69]	Human alpha-synuclein	Brain, *Homo sapiens*	140	−9	35%	1.3	0.8–3.2 µM
Bellmonte et al., 2012 [70]	Hb 98–114	Tick midgut, *Rhipicephalus (Boophilus) microplus*	17	3	52%	−0.62	6.3 µM
Rodríguez et al., 2010 [71]	PgAFP	*Penicillium chrysogenum* RP42C; also found in *Penicillium chrysogenum* Q176	58	4	27%	2.63	0.12–1.0 µM
Gao et al., 2009 [72]	Meucin-18	*Mesobuthus eupeus*	18	2	55%	−0.66	1.9–8.3 µM
Simon et al., 2008 [73]	Human drosomycin-like defensin	Mainly expressed in skin (mRNA), *Homo sapiens*	43	5	25%	3.58	6.25 µM
Cabras et al., 2008 [74]	SP-B	Porcine salivary gland granules	21	1	9%	0.35	58.68 µM
Briolat et al., 2005 [75]	Catestatin	Skin, *Homo sapiens*	21	4	33%	1.98	80 µM
Briolat et al., 2005 [75]	Cateslytin	Chromaffin cells and in secretion medium, bovine	15	5	33%	4.3	10 µM
Landon et al., 2004 [76]	ARD1	*Archaeoprepona demophoon*	41	3	39%	1.6	ND
Kaiserer et al., 2003 [77]	Penicillium antifungal protein	*Penicillium chrysogenum*	55	5	25%	3.12	ND
Lauth et al., 2002 [78]	wb-Moronecidin	Skin/gill, *Morone saxatilis*	23	3	43%	0.38	50–100 µM
Silva et al., 2000 [79]	Gomesin	Hemocytes, *Acanthoscurria gomesiana*	18	6	33%	4.39	ND
Lugardon et al., 2000 [80]	Vasostatin-1	Bovine chromaffin granules, *Bos taurus*	76	−1	35%	2.03	1–10 µM
Gun et al., 1999 [81]	AnAFP	*Aspergillus niger*	58	5	28%	2.42	4–8 µM
Gallo et al., 1997 [82]	Mouse cathelin-related antimicrobial peptide	Adult testis, spleen, stomach, and intestine, mouse, *Mus musculus*	34	6	29%	1.74	100 µM
Lawyer et al., 1996 [83]	Tritrpticin	Synthetic fragment of porcine cathelicidin.	13	4	53%	2.9	ND
Ehret et al., 1996 [84]	Androctonin	*Androctonus australis*	25	8	28%	3.9	25–50 µM
Mor et al., 1994 [85]	Dermaseptin-B2	skin, giant leaf frog, *Phyllomedusa bicolor*, South America	33	4	54%	0.23	125 µg/mL
Mor et al., 1994 [85]	Dermaseptin-S2	Sauvage’s leaf frog, *Phyllomedusa sauvagii,* South America	34	3	52%	−0.14	20 µM
Mor et al., 1994 [86]	Dermaseptin-S3	Sauvage’s leaf frog, *Phyllomedusa sauvagii*, South America	30	6	53%	−0.25	10–20 µM
Mor et al., 1994 [86]	Dermaseptin-S4	Sauvage’s leaf frog, *Phyllomedusa sauvagii,* South America	28	4	71%	−0.91	20–30 µM
Mor et al., 1994 [87]	Skin peptide tyrosine-tyrosine	Skin, the South American arboreal frog *Phyllomedusa bicolor*	36	1	22%	2.69	100 µg/mL
Fehlbaum et al., 1994 [88]	Drosomycin	Fruitfly, *Drosophila melanogaster*	44	1	34%	2.56	6.25 µM
Bellamy et al., 1992 [89]	Lactoferricin B	*Bos taurus*	25	8	48%	2.75	ND
Mor et al., 1991 [90]	Dermaseptin-S1	Sauvage’s leaf frog, *Phyllomedusa sauvagii,* South America	34	3	50%	0.16	30 µM
Wnendt et al., 1990 [91]	Antifungal protein	*Aspergillus giganteus*	51	9	31%	2.1	1 µM
Miller et al., 1989 [92]	Secretory leukocyte protease inhibitor	Tears, saliva, airway, gastrointestines, genital tracts, *Homo sapiens*	107	12	34%	1.87	ND

MIC: minimum inhibitory concentration; ND: no data.

#### 3.1.1. Synthetic Antifungal Peptides

Synthetic and semi-synthetic peptides have garnered attention because some naturally occurring peptides have been associated with low stability and host toxicity [93]. Synthetic AMPs are produced by modifying or combining naturally existing antimicrobial peptides to improve pharmacological properties, reduce side effects, and decrease the immunogenicity of natural AMPs [94]. Some studies have suggested that synthetic peptides are better than natural peptides because they exert antimicrobial activity at lower concentrations compared to the natural AMPs from which they are derived.

Among the most important advantages of designing synthetic peptides from natural sequences is the gain of function since they present activities that are absent in the original model sequence; they reduce allergic response and toxicity due to the fact that during their design, some specific sequences can be suppressed, and their production is less expensive when compared to some purification processes, and the creation is less and less complex due to the existence of many online servers that facilitate the design [95].

Dias et al. [96] found that the synthetic peptides *Rc Alb-PepI* and *Rc Alb-PepII*, based on the primary structure of *Rc -2S-Alb*, exhibited antifungal activity. They showed that *Rc Alb-PepII* inhibited the growth of *Klebsiella pneumoniae* and *Candida parapsilosis*, produced structural alterations on their cell surface, and reduced biofilm formation. On the other hand, it promoted the overproduction of reactive oxygen species capable of oxidizing proteins, DNA, and lipids, which could cause cell death in *Candida parapsilosis*. Finally, in experimental terms, *Rc Alb-PepII* did not generate hemolysis and presented low toxicity in mammalian cells.

Rossignol et al. [97] have shown that the substitution of the amino acid leucine with tryptophan residues in the sequence of a peptide derived from the apolipoprotein E receptor binding region is associated with results such as low cytotoxicity and hemolytic activity; in addition, it increases the spectrum of antifungal activity extended against various *Candida* spp. and early stage *C. albicans* biofilms.

Similarly, it has been shown that the introduction of α, β-dehydro acids, such as α, β-didehydrophenylalanine (ΔPhe), allows the stabilization of the secondary structure and improves resistance to degradation by some enzymes. By evaluating three cationic peptides containing Δphe (IJ2, IJ3, and IJ4), fungicidal activity against yeasts and filamentous fungi were found. The MIC required for such activity ranged from 3.91 to 250 μM; furthermore, the mechanisms of damage were the disruption of cell wall structures and the alteration of membrane permeability, leading to the enhanced entry of the peptide into the cell, the accumulation of reactive oxygen species, and the induction of apoptosis [98].

Concerning synthetic peptides and *A. fumigatus*, Lupetti et al. [30] evaluated the in vitro antifungal activity of the peptides hLF (1–11) and hLF (21–31), dhvar4 and dhvar5, and UBI 18–35 and UBI 29–41, derived from human lactoferrin, ubiquicidin, and histatin 5. The authors found a dose-dependent antifungal activity of all the molecules studied, with dhvar5 showing the best results. With respect to hLF (1–11), dhvar5, and UBI 18–35, it is important to note that they showed effectiveness against *A. fumigatus* conidia. Of the peptides evaluated, only dhvar5 (≥16 μM) and UBI 18–35 (≥20 μM) showed hemolytic activity.

On the other hand, Fioriti et al. [99] evaluated the antifungal activity of two antimicrobial lipopeptides (C14-NleRR-NH_2_ and C14-WRR-NH_2_) against two azole-resistant *A. fumigatus* strains, SSI-4524 and SSI-5586. From the study, they found that both lipopeptides had antifungal activity, with an MIC between 8 mg/L and 16 mg/L. In addition, microscopy showed that hyphal growth was hindered at concentrations at or above the MIC.

#### 3.1.2. Mechanism of Action of Antifungal Peptides

##### Cell Membrane-Targeted Antifungal Peptides

Antifungal peptides have a rapid and broad spectrum of activity in vitro. Although the mechanism of action of antifungal peptides is not widely described, some reportedly bind to nuclear envelope proteins of certain fungi and produce reactive oxygen species and ATP. They may also disrupt membrane surface tension to form pores and release K+ and other ions in the cell [100,101]. Generally, peptides with antifungal activity reported thus far attack the cell membrane, although they can target nucleic acids, organelles, and intracellular macromolecules (Figure 2).

Electrostatic interactions first attract antifungal peptides to the fungal membrane [102]. Subsequently, parallel-oriented AMPs flock to the lipid bilayer due to interactions between hydrophobic residues and the amphipathic structure of the peptide [103]. As their concentration increases, AMPs adopt a perpendicular orientation to the surface, dislocate lipids, and modify membrane structure through electrostatic changes, pore formation, alteration of the permeability barrier, and curvature transformations [56]. These mechanisms are based on widely described models, such as barrel wall, carpet, and annular pore [104,105,106]. Antifungal peptides can target intracytoplasmic structures and inhibit various cellular functions without damaging the membrane, although these mechanisms are poorly characterized. Some studies have reported that AMPs can affect the cell nucleus, inhibit the synthesis of the cell wall and proteins, reduce enzymatic activity, and attack some organelles such as mitochondria, leading to cell death [107,108,109].

On the other hand, AMPs may play an immunomodulatory role because they reduce the levels of proinflammatory cytokines and presumably the probability of developing multiorgan dysfunction during fungal infection [110,111]. Peptides can also promote chemotaxis and the differentiation of macrophages and dendritic cells [112].

##### Cell Wall-Targeted Antifungal Peptides

The antifungal peptides that target the cell wall act on the molecules of importance in the formation of this structure and that play an essential role in the resistance to antifungals; generally, their mechanisms are related to inhibition of B-glucans, the main polysaccharide of the fungal cell wall (50–60% of the dry weight of this structure), formed by glucose fractions joined by glycosidic bonds that form a branched network that confers strength to the cell wall. Another mechanism is the inhibition of chitin synthesis, a component that is synthesized from N-acetyl glucosamine by the enzyme chitin synthase and whose content in the fungal wall depends on the morphological phase of the fungus, reaching 10–20% of the dry weight of the cell wall; it is generally responsible for the rigidity and shape of the cell wall. Finally, another mechanism is mannan-binding, which constitutes the outermost layer of the fungal cell wall and is related to virulence, adhesion, and biofilm formation [113].

Some agents, such as pneumocandin A0, have shown fungicidal activity against pathogens, such as *C. albicans*, but high hemolytic activity and little efficacy against *A. fumigatus*. In contrast, extended-spectrum echinocandins have not only shown fungal activity against *A. fumigatus* but also against Candida species, including those resistant to various conventional antifungals. [114]. On the other hand, although nikkomycin Z reports modest activity against *A. fumigatus*, its combination with echinocandins may improve its efficacy [115].

##### Antifungal Peptides Targeting Intracellular Molecules and Structures

Nucleic acids, organelles, and other fungal macromolecules are not often the target of existing peptides; however, they are a therapeutic target of an increasing number of investigations. Although some antifungal peptides have been shown to bind to DNA, the antimicrobial mechanisms are not completely clear. Recently, a group of authors discovered that inhibiting the protein synthesis and cell replication of pathogenic fungi induces changes in their metabolic pathways [107].

Indolicidin, a peptide isolated from bovine neutrophil cytoplasmic granules, has been associated with significant antifungal activity against *C. albicans, C. krusei*, and *A. flavus*. In addition, the liposomal formulation of this peptide allowed a sufficiently high dosage to successfully treat mice systemically infected with *A. fumigatus* [116,117]. Lee et al. [118] studied the binding of 14-Helical β-Peptides in living fungal cells and artificial membranes; they found that upon entry into the cytoplasm, the peptide is able to rupture the nucleus and vacuoles, leading to cell death. Due to their ability to bind nucleic acids, these peptides behave as antineoplastics and, therefore, can have negative effects on the host due to their high toxicity. Despite the above, this limitation can be counteracted by the use of various formulations, such as nanoparticles and liposomes, which leads to a reduction in adverse effects without eliminating the activity of the compound [113].

#### 3.1.3. Limitations in the Use of Peptides as Antifungals

Like any therapy, antifungal peptides present some limitations that have been progressively documented. These limitations are related to administration, stability, selectivity, toxicity, and possible future resistance.

##### Route of Administration

One variable that is both a limitation and a major challenge in relation to peptides is the route of administration. In their review, Kumar et al. [119] reported that oral administration exposes the peptide to proteolytic digestion by enzymes in the digestive tract, such as trypsin and pepsin. In addition, systemic administration can generate short half-lives in vivo, protease degradation, and cytotoxic profiles in blood. This problem has been partially solved with the postulation of nanoparticles as delivery vehicles.

##### Selectivity and Toxicity

It is important to consider that good in vitro antifungal activity is not sufficient if it is not accompanied by a low toxicity of antifungal peptides to mammals. Some peptides have the ability to specifically target enzymes related to ergosterol or β-glucan synthesis, which translates into high selectivity against the microorganism and a low probability of host cell damage. Additionally, commonly used peptides such as echinocandins have been associated with less liver damage compared to other antifungal agents [120,121].

According to Fernandez et al. [122], there are two essential reasons why antifungal peptides show reduced toxicity in mammals. Firstly, there is a stronger interaction between the fungal membrane characterized as anionic due to the high content of phosphatidylinositol and phosphatidic acid and the cationic charges of the peptide; this contrasts with the mammalian cell membrane, which is predominantly neutral in charge due to the phosphatidylcholine content. Moreover, the antifungal peptides target membrane lipids unique to fungi, which contributes to reduced toxicity in the human host.

##### Peptide Stability

Peptide stability can be compromised due to modifications of variables such as pH, temperature, the action of various proteases, metal ions, chemical reagents, and ultraviolet light. In relation to pH, it has been documented that peptides do not necessarily require neutral conditions since findings have been documented in acidic or alkaline conditions [123]. Additionally, certain ions, such as K ^+^, Na ^+^, Mg ^2+^, Ca ^2+^, among others, also affect the activity of some antifungal peptides [124]. While some authors claim that most antifungal peptides tolerate a maximum of 100 OC, there are reports of sustained activity above 50% after exposure to 121 °C for 30 min and that the activity of the peptides can be sustained for up to 50% after exposure to 121 °C for 30 min [125].

##### Pharmacological Resistance

There have been few findings on the resistance of fungi, specifically *A. fumigatus*, to antifungal peptides. However, a concern for the future is that the increasingly frequent use of antifungal peptides will eventually lead to the emergence of new resistance mechanisms, as has already been documented for conventional antifungals. It is worth noting that, although fungi evolve rapidly, which gives them a great capacity to adapt to hostile environments, the cell membrane (the usual therapeutic target) evolves slowly [122].

## 4. Future Perspectives and Challenges Related to the Use of Antifungal Peptides

Despite the advantages of antifungal peptides in treating fungal infections, unfavorable characteristics, such as poor selectivity, hemolytic activity, and toxicity and instability due to host enzyme degradation, especially among naturally occurring peptides, warrant improvements [56,104]. Although synthetic AMPs have mitigated some issues, there are still several alternatives that can be explored to improve the efficacy and safety of these molecules, including artificial intelligence (AI), lipidation, and the use of nanoparticles as delivery vehicles.

AI algorithms can help develop peptides with activity against multiple pathogenic microorganisms, enabling the production of more effective AMPs with lower costs and time [126]. Neural networks [127], supervised learning [128], random forests [129], and fuzzy clustering [130] are prime candidates among the algorithms used for peptide development. Thus far, AI has been used to generate synthetic peptides and predict antimicrobial activity with quantitative structure–activity relationship (QSAR) models [131]. In 2018, Muller et al. [132] trained a long-term generative memory recurrent neural network (RNN) to recognize different patterns of helical antimicrobial peptide helicases and create novel sequences from them. In their study, they predicted that 82% of de novo sequences would have antimicrobial activity, compared to 65% of randomly sampled sequences with the same amino acid distribution as the training set. Another study [133] used RNNs to produce AMPs whose lengths ranged from 12 to 20 amino acids, and they showed that deep learning techniques can learn the structure of peptides to create new synthetic peptides with antimicrobial activity.

Furthermore, Capecchi et al. [134] trained RNNs and identified eight non-hemolytic molecules that target different bacteria. Otovic et al. [135] recently used a long-term generative memory RNN to engineer a PEP-137 peptide whose administration enhanced the survival rate to 50% in a murine model of *Klebsiella pneumoniae*-induced sepsis.

Notably, fungi are seldom considered among studies that use AI to develop peptides with antimicrobial activity against pathogens. A recent study built a quantitative structure activity relationship model to detect AMPs, and within a single day, the model identified three outstanding AMPs from millions of candidates [136]. On the other hand, Singh et al. [137] used transfer learning to build a classifier that predicts AMPs with an accuracy and precision of 94%, enabling the rapid discovery of new antifungal molecules of natural and animal origin. Moreover, the lipidation of peptides can improve their action. This entails the incorporation of fatty acids, glycophospholipids, and isoprenes at different AMP positions, which increases peptide flexibility and hydrophobicity, as well as interactions with the cell membrane. Finally, solid polymeric and lipidic nanoparticles constructed from natural and synthetic materials, such as cellulose, gelatin, and chitosan have been proposed for the oral delivery of peptides [138]. Natural polymers are especially attractive due to their degradation and faster drug release rate [139,140].

### 4.1. Other Potential Alternatives to Combat A. fumigatus

#### 4.1.1. Vaccines for the Prevention of Aspergillosis

Since their initial development in May 1796, vaccines have advanced medicine by preventing various viral and bacterial diseases. Vaccination has also been proposed for some years as an alternative to combat fungal infections, primarily invasive ones caused by microorganisms such as *Candida* spp., *Cryptococcus* spp., and *Aspergillus* spp., which frequently affect immunocompromised individuals [141,142,143]. Animal models have been used to explore different types of vaccines that could have activity against *A. fumigatus*: panfungal, subunit, crude extracts (fractions derived from cells and fungal culture mediums), and therapeutics [144,145].

Panfungal vaccines use common antigens from different fungal species and even genera to activate the complement system and T-cell immunity to prevent pathogen growth [146]. Alternatively, crude vaccines with either live or killed strains have proven effective in mice; however, this strategy may induce autoimmune responses in humans [147]. Subunit vaccines use recombinant *Aspergillus* spp. proteins whose mechanism is mediated by TCD4+ lymphocytes, and they are associated with prolonged survival in mice [148]. Finally, the therapeutic model proposes allogeneic transplantation of hematopoietic stem cells, during which dendritic cells (DCs) play a major role because they can discriminate between Aspergillus conidia and hyphae in the induction of adaptive TH responses in mice [149].

According to Steven [150], vaccines could benefit patients who are at risk of developing invasive aspergillosis, especially solid organ transplant candidates, bone marrow transplant candidates at the time before or after initial engraftment, patients with myeloid leukemia, and subjects with inflammatory bowel disease prior to the instauration of corticosteroids and tumor necrosis factor blockers.

Despite encouraging results in animals, the implementation of fungal vaccines in humans still poses some challenges. First, many studies are performed in inbred mice given their well-defined immune system and low costs; however, murine and human immune responses greatly differ [151]. On the other hand, some adjuvants proposed in the case of subunit vaccines are toxic and can lead to complications if used routinely in humans [152]. Additionally, an important limitation related to vaccines for *A. fumigatus* is that this pathogen generally infects immunocompromised individuals and could, therefore, worsen an immune disorder after inadequate immunostimulation instead of protective immunity [153].

#### 4.1.2. Nanotechnology to Combat *Aspergillus fumigatus*

New technologies are being developed to improve treatment options for immunocompromised patients with aspergillosis. These technologies include the use of hydrophilic nanoparticles and microspheres to improve drug bioavailability and target the site of infection more effectively. Challenges in developing these technologies include nanoparticle diversity, size dispersion, binding properties, and biophysicochemical properties. Despite these challenges, some progress has been made in the use of nanoparticles and microspheres for antifungal drug delivery. Further research in this field may lead to improved treatments for aspergillosis [154].

Nanoparticles have potential applications in the delivery of antifungal drugs due to their favorable properties, including their small size, multifunctionality, and biocompatibility. Lipid-based nanocarriers are the most studied for this purpose, and many have undergone clinical trials for the management of invasive fungal infections. The commercialization of liposomal amphotericin B has been a significant advancement, allowing for the clinical use of this effective antifungal drug with minimal toxicity. However, amphotericin B is almost the only antifungal drug that has made it to clinical trials and the market in nanoformulations. Therefore, research should focus on overcoming the challenges that hinder clinical translation of nanoparticle-based formulations [155,156].

Nanoparticles (NPs) are particles with a diameter in the range of 1–1000 nm. These particles have different chemical, physical, or biological properties than their larger counterparts, making them useful for drug delivery applications. NPs used in drug delivery can be broadly classified into phospholipid vesicles (e.g., liposomes), non-phospholipid vesicles (e.g., niosomes), polymeric NPs, polymeric micelles, solid lipid nanoparticles, nanostructured lipid carriers, nanoemulsions, and dendrimers. These different types of NPs have unique properties and potential applications in drug delivery [155,157]. One of the most studied nanoparticle carriers for antifungal drug delivery is liposomes, which have been successful in clinical trials for the management of invasive mycoses.

Liposomes are one type of nanoparticle that have been studied for this purpose and have been successfully used to deliver amphotericin B, which has been shown to be effective in treating systemic fungal infections. Nystatin, another antifungal agent, has also been successfully delivered using liposomes and was shown to be as active or more active than the free drug in vitro. However, further studies are needed to explore the clinical translation of these nanoparticle-based formulations. Other nanoparticle systems, such as polymeric nanoparticles and dendrimers, have also been studied for antifungal drug delivery with promising results [155,158]. Nanoliposomes are a form of drug delivery that are gaining popularity due to their safety, patient compliance, high interlocking efficiency, and rapid action. Several of the biological effects of natural essential oils, including fungal inhibition, are of great interest [159].

The nanoliposome study by Hassanpour et al. [160] showed that the liposomal formulation of voriconazole had a greater inhibitory effect on the growth of fluconazole-resistant *C. albicans* strains compared to the use of voriconazole alone. In addition, the liposomal formulation of voriconazole reduced the expression of azole-resistant genes compared with the use of voriconazole alone. These results suggest that the liposomal formulation of voriconazole could be an effective option for treating *C. albicans* infection in cases of fluconazole resistance.

Ethosomes and transethosomes are types of vesicles that are used as drug carriers and have the advantage of improving penetration through the skin. Ethosomes and transethosomes are also emerging as potential carriers for antifungal drugs. Ethosomes are soft vesicles that are used to improve drug penetration through the skin. They are composed mainly of phospholipids, ethanol, and water. The ability to control the ethanol content in ethosomes allows their size to be regulated, eliminating the need for sophisticated equipment. In addition, the presence of ethanol in ethosomes confers a negative charge that enhances their colloidal stability. The improved skin penetration is due to the ability of ethanol to fluidize ethosomal lipids and the intercellular lipid of the stratum corneum. Transethosomes, in addition, contain an edge activator that enhances permeation even further; however, there is limited information due to the fact that they are relatively new [155,161,162]. Another vesicle of the non-phospholipid type are niosomes, which are used to deliver drugs more efficiently in the body. They are composed of nonionic surfactants instead of phospholipids, which allow them to improve their chemical stability and increase their drug-loading capacity. In addition, they have a lower price and can be stored under normal conditions. However, their physical stability can be affected by the melting and aggregation of particles [155,163].

Polymeric nanoparticles are biopolymers of natural or synthetic origin and are suitable for encapsulating lipophilic drugs and have revolutionized the field of drug delivery, particularly cancer chemotherapy. Their nanometer size allows the drug to permeate cells and effectively destroy the organism. These nanoparticles have demonstrated an excellent ability to enhance the therapeutic properties of drugs while minimizing their side effects and toxicity. Many polymeric nanoparticle formulations based on cytotoxic drugs are already available in the clinical market and many others are under development [155,158]. One class of nanoparticles are solid lipid nanoparticles (SLNs) and nanostructured lipid carriers that improve lipid permeability and stability and allow the co-delivery of several drugs. However, they have the disadvantage of low drug loading capacity and drug expulsion during storage [155,164].

Polymeric micelles are a type of nanoparticle used in drug delivery. These nanostructures are self-assembled from amphiphilic copolymers in water. Polymeric micelles have greater stability against dilution than surfactant micelles. In addition, they can incorporate hydrophobic drugs in their core, which improves their solubility in water. Polymeric micelles are also biodegradable and biocompatible, which limits immune reactions in vivo [155,165].

Nanoemulsions are a type of drug delivery system that is composed of an isotropic mixture of drugs, lipids, and surfactants with small droplet diameters. This formulation has a high drug solubilization capacity and good skin penetration, making it suitable for use in the treatment of fungal infections. In addition, nanoemulsions can be used as an alternative to less stable lipid carriers, such as liposomes. The targeted topical delivery of antifungal drugs via nanoemulsions can maximize the local effects of the drug and avoid systemic toxicity [166].

The use of nanometals is also presented as an antifungal alternative against Aspergillus; in the work of Yu et al. [167], the antifungal efficacy of nanometals (Ag, Cu, and Ni) supported as catalysts was investigated. Most of the previous studies focused on the bactericidal efficacy of nanometals. However, it is important to also investigate the antifungal efficacy because molds and their spores are more resistant than bacteria and can accumulate in high concentrations in humid environments. The critical Ag concentration to inhibit the germination and growth of *A. niger* spores was found to be 65 mg/mL for a 5% wt% nano Ag catalyst, which is lower than several cases in previous studies. In addition, ozone was found to have a synergistic effect on the antifungal efficacy of nanometals. TiO_2_ catalysts loaded with nano-Ag and -Cu were shown to be effective in reducing the survival ratio of *A. niger* spores in the dark. The results of this study may be useful in developing new ways to combat fungi and shows the importance of using silver nanoparticles (Ag NPs) ranging from 1 to 100 nm in size, and their diverse medical applications enable activity across many targets [168]. The particle first binds to the fungal membrane, modifying its permeability and altering cell viability; in addition, it can compromise the respiratory capacity and stop cell division, causing cell death [168]. Nanoparticles inactivate enzymes by releasing Ag ions from thiol groups; additionally, smaller particles and positively charged functional groups that bind to the protein corona of silver nanoparticles induce cell toxicity [157].

Although nanometals are a promising alternative to treat infections caused by *A. fumigatus*, they are not without limitations such as the raw material from which they are obtained, the production method, biodistribution, and probable toxic effects in humans [168]. Preliminary studies have shown that, due to their size, Ag NPs can cross the cell membrane and, therefore, generate oxidative stress, impair mitochondrial function, and cause cell death [169]. Nanoparticles can also cross the blood–brain barrier and accumulate in the central nervous system [170].

Another study with nanometals was realized by Almansob et al. [164]; this study evaluated the use of gold nanoparticles (AuNPs) synthesized from an extract of *Mentha piperita* to treat fungal infections. The results showed that these nanoparticles did not have a particularly effective antifungal effect against multidrug-resistant species of Aspergillus. However, inhibition was observed in five of the isolates tested, and there were significant changes in the extracellular enzyme activity of nosocomial fungi treated with gold nanoparticles.

## 5. Conclusions

In conclusion, antifungal peptides are a promising therapeutic alternative for the treatment of fungal infections since they have a broad spectrum of activity and can overcome the limitations of conventional antifungal agents. However, the use of these peptides is limited by the properties of these molecules, such as their instability and toxicity, as well as their lack of selectivity. In addition, the emergence of drug resistance and the need for improved delivery systems are major challenges for the development of antifungal peptides. The use of artificial intelligence, lipidation, and nanoparticles as delivery vehicles may help to overcome these challenges and improve the efficacy of these molecules. In addition, other potential alternatives, such as vaccines and nanotechnology, are being studied as potential treatments for aspergillosis.

## Figures and Tables

**Figure 1 jof-09-00042-f001:**
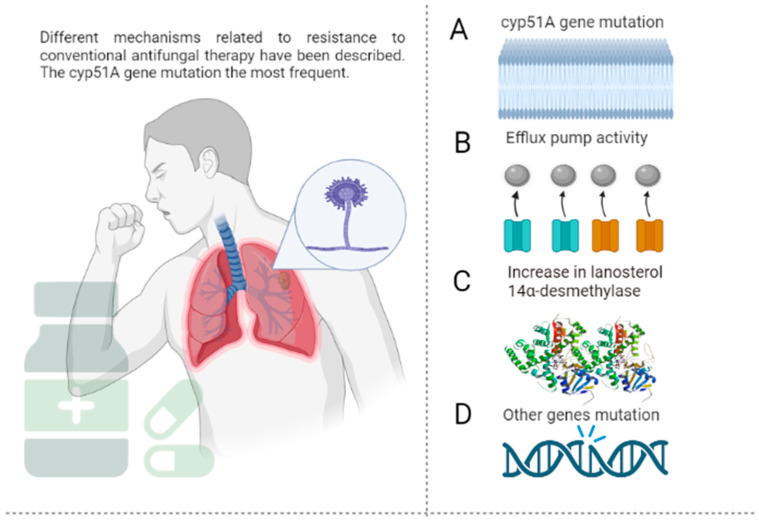
Mechanisms of resistance of *A. fumigatus* to conventional antifungals. (**A**)—Resistance to azole antifungals is often derived from the mutation of the *cyp51A* gene, which codes lanosterol 14-α-demethylase, an important enzyme in ergosterol synthesis. More than 30 mutations have been identified, including the amino acid substitution Gly54, Pro216, Phe219, Met220, and Gly448. Resistance-associated loss-of-function mutations of *ERG3* protect fungal cells from damage by the toxic 14α-methyl-3,6-diol product due to the accumulation of 14α-methylfecosterol that replaces ergosterol and leads to functional membranes, negating the action of azoles in the ergosterol biosynthetic pathway. (**B**)—The efflux pumps deliver the drug to the extracellular space, ensuring a lower concentration at the target site. This action is mediated by some protein superfamilies such as the ATP-binding cassette (ABC) and the major facilitator superfamily (MFS). (**C**)—Overexpression of *ERG11* results in increased concentrations of lanosterol 14-α-demethylase and, consequently, higher amounts of the antifungal are required to inhibit the enzyme. (**D**)—Mutations different from *cyp51A*, such as *cyp51B* (which shares 59% of the *cyp51A* sequence), are less frequent, and their implications for azole resistance have not been extensively studied. The figure was created with https://app.biorender.com.

**Figure 2 jof-09-00042-f002:**
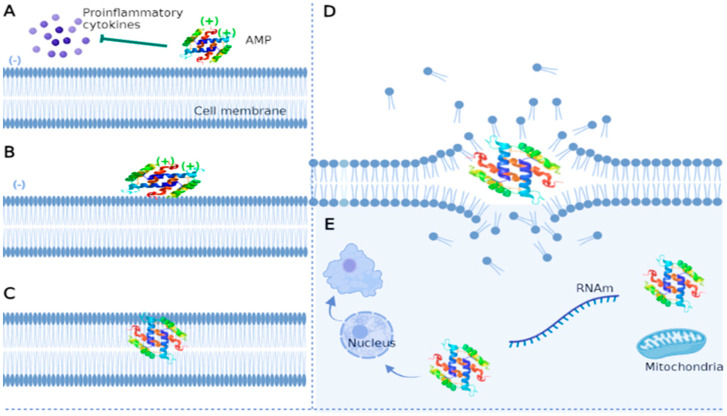
Main mechanisms of action of antifungal peptides. (**A**)—AMPs can play immunomodulatory roles by inhibiting the production of proinflammatory cytokines such as IL-1, IL-6, and tumor necrosis factor. (**B**)—AMPs interact with the fungal membrane through electrostatic interactions due to charge differences (negatively charged membrane and positively charged peptide). (**C**)—The hydrophobic character of AMP enables its insertion into the membrane through a perpendicular orientation as its concentration increases. (**D**)—AMPs dislocate lipids and destroy the membrane. (**E**)—The peptide can enter the cell and damage various structures such as the nucleus, inhibit RNA synthesis, attack mitochondria, and induce functional alterations up to cell death. The figure was created with https://app.biorender.com.

## Data Availability

Not applicable.

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
