# Peer review of "Specific Focus on Antifungal Peptides against Azole Resistant Aspergillus fumigatus: Current Status, Challenges, and Future Perspectives"

_jof, 2022, doi:10.3390/jof9010042_

Round 1

Reviewer 1 Report

The authors of the manuscript "Antifungal peptides, a therapeutic alternative against Aspergillus fumigatus resistant to conventional antifungals", concentrate on the evidence of various peptides with antifungal activity, focusing the document on the control of resistant fungal strains of A. fumigatus, as well as the use of some alternative therapies such as vaccines and nanoparticles.

In that sense, this review requires major adjustments such as:

Reorder the topics according to the order presented in the abstract; at the end of the manuscript, discuss the topics of vaccines and nanoparticles (alternative therapies) since they are not the central axis of the review.

The images presented are not adequate. It is suggested to place a figure that illustrates the resistance mechanisms of A. fumigatus and the mechanisms of action of the peptides (in Fig. 2, they only place the mechanism of action); showing the reader the advances in research on antifungal peptides due to the variety of review articles on this topic:

1. Antifungal Peptides as Therapeutic Agents https://doi.org/10.3389/fcimb.2020.00105

2. Antifungal peptides: potential candidates for the treatment of fungal infections. https://doi.org/10.1517/13543784.9.2.273

3. Antimicrobial peptides: A new frontier in antifungal therapy. https://doi.org/10.1128/mBio.02123-20

Eliminate or modify figure 1; it is not relevant. Of the three therapeutic alternatives for patients with A. fumigatus infection, the studies with vaccines and nanoparticles are in tests with mouse models and cell lines, respectively. In contrast, some peptides are available to the patient or in preclinical trials.

Delete the sequence column from Table 1.

Carry out a detailed review of the literature consulted and cited; more than 20% are not indexed (without impact factor).

Take care of the details; for example, they refer to the activity of the ZnO NPs by the reference of Rajeshkumar et al [60], “On the other hand, Rajeshkumar et al [60] found that ZnO NPs exhibit pronounced antimicrobial activity against pathogenic fungi including A. fumigatus. Additionally, they evidenced that the area of inhibition increases with concentration, from 11.17 ± 0.15 mm at 50 µL volume to 13.10 ± 0.13 mm at 150 µL volume”. However, Rajeshkumar reports the activity of silver NPs and not of ZnO NPs. In addition, in the cited article, the images of antifungal activity generate speculation about the inhibition halos, so each citation must be carefully selected. These errors leave in doubt the interpretation and quality of the cited material.

In the section on NPs as use for patients with A. fumigatus infection, more information and other references should be included regarding other nanoparticles and not only ZnO and Ag, such as https://doi.org/10.1002/btpr. 3206; https://doi.org/10.3390/nano12050814. In addition, in the text, they must make it clear if the synthesis of nanoparticles is chemical, mechanical, or biological.

If you are referring to multiple species of the same genus, “spp.” can be used. These abbreviations should not be italicized. (https://besjournals.onlinelibrary.wiley.com/doi/10.1111/2041-210X.12594)

Author Response

We thank the reviewer of the manuscript for his valuable comments and below we show the responses to what was requested:

1.Reorder the topics according to the order presented in the abstract; at the end of the manuscript, discuss the topics of vaccines and nanoparticles (alternative therapies) as they are not the main focus of the review.

Response: taking into account the valid request made by the reviewer, the manuscript is reordered giving more importance to the central theme which is antifungal peptides. Vaccines and nanoparticles are left at the end of the document in section 4.1 under the heading "other potential alternatives".

2.The images presented are not adequate. It is suggested to place a figure illustrating the mechanisms of resistance of A. fumigatus and the mechanisms of action of the peptides (in Fig. 2, only the mechanism of action is placed); showing the reader the advances in the research of antifungal peptides due to the variety of review articles on this topic:

Response: based on the reviewer's suggestion, it is decided to elaborate a figure illustrating the main mechanisms of resistance to antifungals (Section 2, Figure 1). Additionally, Figure 2 is left to illustrate the main mechanism of action of MAPs, and the mechanism of attack to other therapeutic targets is described based on the references proposed by the reviewer.

  1. Delete or modify Figure 1; it is not relevant. Of the three therapeutic alternatives for patients with A. fumigatus infection, studies with vaccines and nanoparticles are in trials with mouse models and cell lines, respectively. In contrast, some peptides are available to the patient or in preclinical trials.

Response: Recognizing the low relevance of Figure 1 submitted in the first manuscript, it is decided to delete as suggested by the reviewer.

  1. Remove the sequence column from Table 1.

Response: The sequence column of Table 1 is eliminated.

5.Perform a detailed review of the literature consulted and cited; more than 20% are not indexed (no impact factor).

Response: The suggested review was carried out by limiting the number of articles published in journals without impact factor.

  1. In the section on NPs as a use for patients with A. fumigatus infection, more information and other references on other nanoparticles and not only ZnO and Ag should be included. In addition, in the text it should be made clear whether the synthesis of nanoparticles is chemical, mechanical or biological.

Response: The information in this segment is expanded by mentioning other nanoparticles with therapeutic potential against A. fumigatus. The studies suggested by the reviewer are considered (see section 4.1.2).

Reviewer 2 Report

The review paper by Pimienta and colleagues supplies an interesting view into alternative therapeutic potentials against Aspergillus fumigatus infections. The paper clearly describes the current status of treatment, drawbacks and alternatives and has additional value for the experts in the field. However, before the paper is acceptable for publication in Journal of Fungi, a number of aspects needs to be addressed.

1.       Journal of Fungi has an international audience in which English is the preferred language. At least 16 references point to papers written in Spanish. Although there are no doubts about the quality of the referred papers, these references must be omitted from the manuscript and replaced by alternatives written in English.

2.       References to websites must be omitted from the reference list and inserted into the text.

3.       The text in lines 270-275 is not clear and needs some more explanation. What is meant by non-ribosomal proteins?

4.       Table 1 is hardly explained. The legend to this table should be elaborated. The table now contains antifungal peptides from natural sources. What is known about synthetic peptides with similar activities?

5.       During the last decades, a huge amount of data is released about extracellular vesicles from fungi and infected hosts and their roles in host immunity. Since these extracellular vesicles may also deliver short peptides, a brief description must be included in the manuscript.

6.       A major drawback of the application of antifungal peptides might be the administration of these potential antifungal drugs. So, also this item should be addressed in the manuscript.

Author Response

We thank the reviewer of the manuscript for his valuable comments and below we show the responses to what was requested:

1.Journal of Fungi has an international audience with English being the preferred language. At least 16 references point to papers written in Spanish. Although there is no doubt about the quality of the referenced papers, these references should be omitted from the manuscript and replaced by alternatives written in English.

Response: Considering the primary audience of the journal, the Spanish language references are eliminated.

  1. References to web pages should be omitted from the list of references and inserted in the text.

Answer: the reviewer's suggestion is followed.

  1. The text of lines 270-275 is not clear and needs some further explanation. What is meant by non-ribosomal proteins?

Answer: the reviewer's suggestion is followed.

  1. Table 1 is hardly explained. The legend of this table should be elaborated.

Answer: the reviewer's suggestion is followed.

  1. The table now contains antifungal peptides from natural sources. What is known about synthetic peptides with similar activities?

Answer: taking into account the reviewer's suggestion, section 3.1.1 is added in which synthetic peptides are discussed. 

  1. One of the main drawbacks of the application of antifungal peptides could be the administration of these potential antifungal drugs. Therefore, also this point should be addressed in the manuscript.

Response: taking into account the reviewer's suggestion, section 3.1.3 discussing potential problems and limitations of peptides with antifungal activity is added.

Reviewer 3 Report

In this review, Pimienta et al. present some information on antimicrobial peptides (AMPs) demonstrating antifungal activity as candidates

My major comment to this work is that this is poorly balanced. Surprisingly, only less than 50% of the body text  (about 3 pages + Table 1) is truly devoted to AMPs. This proportion should be changed. Part 2, parts 3.1 and 3.2 should be substantially shortened, while 3.3, especially 3.3.1 could be expanded. More details on postulated mechanisms of antifungal action of AMPs should be provided. Since the authors consider AMPs as good candidates for antifungals potentially  useful for the treatment of aspergilloses, a very important aspect of molecular basis of their possible selective toxicity is almost completely absent. Good antifungal in vitro activity is not enough and must be accompanied by low mammalian toxicity. For example, in the case of AMPs disturbing fungal cell membrane, it is crucial to know if this effect is specific for the fungal, not mammalian membrane. The same is true for antifungal AMPs with postulated other mechanisms of antifungal action.

Therefore, I would suggest that a substantial rewriting of this ms. is necesssary before its final acceptance.

Author Response

We thank the reviewer of the manuscript for his valuable comments and below we show the responses to what was requested:

1.My main comment on this paper is that it is poorly balanced. Surprisingly, only less than 50% of the body of the text (about 3 pages + Table 1) is actually devoted to MPAs. This proportion should be changed.

Response: Based on the reviewer's comments, it is decided to expand the information on antifungal peptides. New sections are added ( 3.1.1 and 3.1.3); in addition, the information in existing sections is expanded ( 3.1.2).

  1. Parts 3.1 and 3.2 of Part 2 should be substantially shortened, while 3.3, especially 3.3.1, could be expanded. More details should be provided on the postulated mechanisms of antifungal action of AMPs.

Response: The reviewer's suggestions are followed, considering also the comments of the two additional reviewers.

  1. Since the authors consider MPAs to be good candidates for potentially useful antifungals for the treatment of aspergillosis, one very important aspect is almost completely missing: the molecular basis of their possible selective toxicity. Good in vitro antifungal activity is not sufficient and must be accompanied by low mammalian toxicity. For example, in the case of AMPs that alter the fungal cell membrane, it is crucial to know whether this effect is specific to the fungal membrane and not to the mammalian membrane. The same is true for antifungal AMPs with other postulated antifungal mechanisms of action.

Response: Taking into account the reviewer's suggestion, section 3.1.3 is added, in which the possible problems and limitations of peptides are discussed.

Round 2

Reviewer 1 Report

The manuscript was submitted carelessly; it is difficult to review, and the sequence is lost. Please resubmit the document highlighting only the changes, removing the text removal, formatting corrections, and the sections that have in Spanish

Author Response

Thank you for your comments. We did the changes.

Reviewer 3 Report

The authors appropriatelly addressed all comments of the reviewer. I would recommend acceptance of this ms. for publication in JoF after some language polishing.

Author Response

Thank you for your comments.

Round 3

Reviewer 1 Report

The authors have made all the suggested comments; however they must review the English and their bibliographical references before publication

Author Response

Thank you for your time in helping us improve the manuscript. We attach the paper with grammar corrections, use of italics in species, and added author contributions and funding.
